# High-Performance Extraction Process of Anthocyanins from Jussara (*Euterpe edulis*) Using Deep Eutectic Solvents

**Nicholas Vannuchi [1], Anna Rafaela Cavalcate Braga [2] and Veridiana Vera De Rosso [3,*]**

1. Programa de Pós-Graduação Interdisciplinar em Ciências da Saúde, Universidade Federal de São Paulo (UNIFESP), Rua Silva Jardim 136 CEP, Santos 11015-020, São Paulo, Brazil; nickvannuchi@gmail.com
2. Department of Chemical Engineering, Campus Diadema, Universidade Federal de São Paulo (UNIFESP), Rua São Nicolau 201 CEP, Diadema 09972-270, São Paulo, Brazil; ana.braga@unifesp.br
3. Nutrition and Food Service Research Center, Universidade Federal de São Paulo (UNIFESP), Rua Silva Jardim 136 CEP, Santos 11015-020, São Paulo, Brazil
* Correspondence: veridiana.rosso@unifesp.br; Tel.: +55-11-99658-3459

**Abstract:** New strategies for obtaining target bioactive compounds and natural pigments with the use of "green solvents" are consistently being developed, and deep eutectic solvents are (DES) a great alternative. This work established the significant variables and models for anthocyanin extraction, using DES and experimental design, of *Euterpe edulis* Mart. (jussara) fruit pulp, an endangered palm tree from the Brazilian Atlantic Forest. From a screening of seven initially tested DES, choline chloride/xylitol-based solvents had the best results with up to 42% increase in the total anthocyanin yield compared to methanolic extraction. Antioxidant assays also revealed a maximum antioxidant capacity of 198.93 mmol Trolox/100 g dry weight basis. The DES extract showed slower degradation to heat at 60° and 90 °C (2.5 times) and indoor constant light source (1.9 times) than methanolic extracts. The optimal extract also revealed slight inhibition of *S. enterica* and *S. aureus* growth in the agar plate.

**Keywords:** deep eutectic solvent; sustainable chemistry; antioxidant activity; *Euterpe edulis*; experimental design

## 1. Introduction

Volatile Organic Solvents (VOS) are estimated to account for 60% of all industrial emissions and 30% of all volatile organic compounds emitted worldwide [1,2]. Although commonly used to extract organic compounds, such solvents cause environmental and social damage due to their limitations on reuse, toxicity, flammability, volatility, and presence of residues in the final product [3–5]. These risks have led to restrictive legislation such as Directive 2010/75/EU proposed by the European Parliament, which aims to limit the emission of certain VOS by industries.

Seeking to work around this problem, new technologies aiming at safer and more sustainable alternatives for VOS use have emerged, among them the so-called deep eutectic solvents (DES). These liquids have more desirable solvent characteristics such as negligible volatility, non-flammability, chemical, thermal, and electrochemical stability, and the ability to be synthesized from non-petroleum derived organic and/or inorganic ionic species [6]. Considering their unique properties and attractive price, DES have become a promising topic for academic research and industry for both separation processes and extraction of plant bioactive compounds [7,8].

Anthocyanins are a class of natural pigments with underexplored use in the food and cosmetics industry, which commonly uses artificial colorants. DES have been successfully tested for anthocyanin extraction, contributing to the "clean label" of the product, a strategy used by industry to attract increasingly demanding consumers seeking environmentally responsible consumption and healthy eating choices [7,9]. Potential health risks involving

artificial colorants, including allergic reactions and carcinogenic evidence, particularly of blue dyes [10,11], among other risks, have led to the banning of various artificial colorants in various countries [11,12].

Based on the above, DES are ideal candidates for developing new techniques for the extraction and preparation of poorly explored fruits, like the jussara (*Euterpe edulis* Mart.). This palm tree from the *Arecaceae* family has a large range of distribution, residing mostly along the Brazilian coastal Atlantic Forest, with occurrences from northeast Brazil to eastern Paraguay and northern Argentina. The tree is an important food source for many birds and mammals, who work as seed dispersers [13]. Despite the large area that it has occupied, jussara is at risk of extinction because of deforestation, defaunation, and clandestine heart of palm extraction (which kills the plant), and in some cases, it has even been locally eliminated [14]. Alternative sustainable use of the palm include the preparation of frozen pulp from the fruit, employed by small communities that live near the forest and consumed locally in sorbets, juices, and children's snacks [15,16]. Sustainable use of the plant by collecting its berries is an alternative of great economic and environmental potential, which is being pursued by these communities, mostly in southern states of Brazil, where they use the discarded jussara seeds to aid reforestation [17,18].

The pulp has been studied recently, with most studies evaluating the composition and different methods of extracting the phenolic and anthocyanin content. Large concentrations of anthocyanins and other antioxidant compounds, high lipid content and good quantities of minerals such as potassium, zinc, copper, and cobalt were identified, increasing the interest for the fruit [19,20]. Some studies regarding extraction methods to obtain anthocyanins and phenolic were conducted [21–23], but mostly using methanol and ethanol, with the best results coming from ultrasound assisted extraction [16].

This study proposes to develop processes for obtaining anthocyanins from jussara pulp, using DES and water as co-solvents for the first time, providing an alternative and exploration technique for the fruit through green extraction methods, and optimizing the process using experimental design. The resulting extract of concentrated anthocyanins presents a new technology for fruit usage, as a natural colorant, further encouraging their exploration over the heart of palm extraction.

## 2. Material and Methods

### 2.1. Fruit Gathering

The *Euterpe edulis* fruits were collected in São Paulo, Brazil (Ubatuba city: 23°19′01.47″ S, 44°52′37″ W) in January 2014 by Instituto de Permacultura e Ecovilas da Mata-atlântica (IPEMA). All fruits were sanitized with running water. The fruit was depulped in specific equipment used to separate jussara pulp from seed and peel (the same used by açaí), aided by warm water (45 °C maximum, to protect the seed and the anthocyanins) resulting in a thick purple juice. This juice is then immediately frozen at −40 °C, lyophilized for 48 h, and then stored at −40 °C to preserve the anthocyanin content. The moisture content was performed according to the methods of AOAC [24].

### 2.2. Eutectic Solvent Synthesis

The eutectic solvents were synthesized by a heating method [25]. Compounds used in the study are described in Table 1, based on the literature [25–28]. They are Chlorine Chloride (Ch), L-proline (Lp), Levulinic acid (Leu), Butene-1,4-diol (but), and Glycerol (Gly), Xylitol (Xyl). The DES mixtures were stirred at 80 °C for 1 h until a clear liquid was obtained with minimum addition of water. They were subsequently frozen at −40 °C and then lyophilized to ensure removal of water. A total of seven mixtures of DES were prepared using methods and molarities already described in the literature as follows: [Ch-Leu 1:2], [Ch-But 1:2], [Ch-Gly 1:2], [Ch-Xyl 5:2], [Lp-Leu 1;2], [Lp-But 2:5], and [Lp-Gly 2:5].

**Table 1.** Properties of the compounds used to create deep eutectic solvents.

| Compound | Chemical Structure | Molar Mass (g/mol) | Density (g/mL) at 20 °C | Chemical Characteristic |
|---|---|---|---|---|
| 1-5 Chorine chloride | | 139.62 | 1.10 (aq) | Hydrogen bond acceptor |
| L-proline | | 115.13 | 1.35 | Hydrogen bond acceptor |
| Levulinic acid | | 116.116 | 1.14 | Hydrogen bond donor |
| Butane-1,4-diol | | 90.121 | 1.02 | Hydrogen bond donor |
| Glycerol | | 92.0776 | 1.26 | Hydrogen bond donor |
| Xylitol | | 152.12 | 0.77 | Hydrogen bond donor |

*2.3. Experimental Design: Response Surface Methodology*

We evaluated the seven synthetized DES potentials for extracting anthocyanins by ultrasound-assisted extraction (Eco-sonics Ultronique, Campinas, Brazil), with fixed potency of 400 W for all experiments. An initial screening to choose 2 of the seven initial DES was conducted comparing anthocyanin content in the DES extracts and control (MeOH) with 30% co-solvent (acidified water), 1:15 sample:solvent ratio, and 3 extraction repetitions, during 5 min in the ultrasonic probe. Two of the seven initial DES were chosen (DES1 and DES2), based on anthocyanin yield, then two experimental designs were conducted: first, a fractional factorial experimental design $2^{5-1}$ (FFED) to evaluate possible non-significant variables, and second, a central composite rotatable design (CCRD) with remaining significant variables. The FFED $2^{5-1}$ with three central points and three levels for each independent variable was adopted, giving a total of 19 trials (Table 2 and S1A). The independent variables evaluated for each extraction were the following: (i) eutectic solvent composition; (ii) solid–liquid ratio ($R_{(S/L)}$), meaning the mass (g) of fruit pulp per mass of solution (g), (iii) number of extraction repetitions, (iv) time of extraction, and (v) co-solvent content (%) were chosen. The co-solvent (water) was acidified with HCl 3% to ensure the protonated state of anthocyanins [29]. Methanol with the same co-solvent proportion and parameters was used as a control. After the extractions, the extracts were filtered with filter paper (Unifil qualitative 80 g/m$^2$) and under a vacuum, then concentrated using a rotary evaporator (<37 °C), removing the water content of the co-solvent for quantification. The results were evaluated using ANOVA followed by Dunnett's test considering a confidence level of 95%, comparing the DES results to the control (MeOH extractions) in terms of total anthocyanin level (TAL). The response was expressed in $\mu g_{anthocyanins}/mL$.

An estimate of the main effect was obtained by evaluating the difference in the total anthocyanin yield caused by a change from the low (−1) to the high (+1) levels of the corresponding variable. The fractional factorial experimental design's focus was to discover the significant variables and their main interactions with the total yield of extracted anthocyanins. Therefore, after analyzing the data, the variables that had a significative effect on the total yield of the process were fixed and the others removed (including the removal of DES2 from further trials) for an additional assay CCRD $2^2$ plus axial and central

composites, with three replicates at the central point, giving a total of 11 additional trials. Table 3 shows the real values (coded values in Table S1B) and levels used at CCRD $2^2$. After analyzing the CCRD results, the best conditions for the anthocyanin extraction with the best DES was determined, and the model was validated in triplicate to be used in further trials.

**Table 2.** Independent variable values of fractional factorial experimental design ($2^{5-1}$) with three central points for DES1 and DES2 were used as independent variables as well as $R_{(S/L)}$, Repetitions, time, and $R_{(CoSolvt\%)}$.

| Assay | DES | $R_{(S/L)}$ | Extraction Repetitions | Time (min) | $R_{(CoSolvt\%)}$ |
|---|---|---|---|---|---|
| | $X_1$ | $X_2$ | $X_3$ | $X_4$ | $X_5$ |
| 1 | DES1 | 1:15 | 1 | 2 | 40 |
| 2 | DES2 | 1:15 | 1 | 2 | 20 |
| 3 | DES1 | 1:25 | 1 | 2 | 20 |
| 4 | DES2 | 1:25 | 1 | 2 | 40 |
| 5 | DES1 | 1:15 | 5 | 2 | 20 |
| 6 | DES2 | 1:15 | 5 | 2 | 40 |
| 7 | DES1 | 1:25 | 5 | 2 | 40 |
| 8 | DES2 | 1:25 | 5 | 2 | 20 |
| 9 | DES1 | 1:15 | 1 | 6 | 20 |
| 10 | DES2 | 1:15 | 1 | 6 | 40 |
| 11 | DES1 | 1:25 | 1 | 6 | 40 |
| 12 | DES2 | 1:25 | 1 | 6 | 20 |
| 13 | DES1 | 1:15 | 5 | 6 | 40 |
| 14 | DES2 | 1:15 | 5 | 6 | 20 |
| 15 | DES1 | 1:25 | 5 | 6 | 20 |
| 16 | DES2 | 1:25 | 5 | 6 | 40 |
| 17 | DES1/DES2 | 1:20 | 3 | 4 | 30 |
| 18 | DES1/DES2 | 1:20 | 3 | 4 | 30 |
| 19 | DES1/DES2 | 1:20 | 3 | 4 | 30 |

**Table 3.** Independent variable values used in CCRD $2^2$ assays with 3 central points, used as independent variables R (CoSolvt%) and extraction repetitions.

| Assay | R (CoSolvt%) | Extraction Repetitions |
|---|---|---|
| | $X_1$ | $X_2$ |
| 1 | 16 | 2 |
| 2 | 44 | 2 |
| 3 | 16 | 6 |
| 4 | 44 | 6 |
| 5 | 10.3 | 4 |
| 6 | 49.7 | 4 |
| 7 | 30 | 1 |
| 8 | 30 | 7 |
| 9 | 30 | 4 |
| 10 | 30 | 4 |
| 11 | 30 | 4 |

The main response evaluated for optimization of the process was based on the total yield of cyanidin 3-glucoside (C3G), cyanidin 3-rutinoside (C3R), and total anthocyanin level (TAL), considering only these two major anthocyanins, attempting to obtain models for those responses as well. Statistica 14.0 software was used to analyze the results and plot the response surfaces.

### 2.4. Identification and Quantification of Anthocyanins

For all the samples, anthocyanin separation was carried out on a 250 × 4.6 i.d. mm, 5 µm particle size, $C_{18}$ Shim-pack CLC-ODS column (Shimadzu, Canby, OR, USA), using as the mobile phase a gradient of methanol/5% formic acid (*v/v*) from 15:85 to 80:20 in 25 min,

the latter proportion being maintained for a further 15 min, at a flow rate of 0.9 mL/min and a column temperature set at 28 °C. The chromatograms were processed at 280 and 520 nm, and the spectra were obtained between 250 and 800 nm [19,30]. The anthocyanins were identified based on the combined information provided by the elution order in the reversed phase column, co-chromatography with standards, UV–VIS, and mass spectra compared to the literature data. Anthocyanins were quantified by HPLC as cyanidin 3-glucoside and cyanidin 3-rutinoside, using an external calibration curve for both with a minimum of six concentration levels. All analyses were performed in triplicate.

### 2.5. Antioxidant Activity

In order to compare the antioxidant activity of extracts obtained by DES in CCRD trials, ABTS [2,2′-azinobis-(3-ethylbenzothiazoline-6-sulfonate)] assay was performed as described by [31]. Antioxidant activity (AA) of the extracts was expressed in μmol Trolox/g equivalents. Results were then plotted in CCRD again for the same effects, using antioxidant activity as the dependent variable and analyzed by ANOVA with a 90% confidence level.

The oxygen radical absorbance capacity (ORAC) method was also performed as previously described by [32], which consists of measuring the decrease in the fluorescence of a protein as a result of the loss of its conformation when it suffers oxidative damage caused by a source of peroxyl radicals. The measurements were taken in triplicate. The ORAC values, expressed as μM Trolox equivalents (μM TE), were calculated by applying the following Equation (1):

$$ORAC(\mu M\ TE) = \frac{C_{trolox} \cdot \left(AUC_{sample} - AUC_{blank}\right) \cdot k}{\left(AUC_{trolox} - AUC_{blank}\right)} \tag{1}$$

where $C_{Trolox}$ is the concentration (μM) of Trolox (20 μM), k is the sample dilution factor, and AUC is the area below the fluorescence decay curve of the sample, blank, and Trolox, respectively. Results were then plotted in CCRD again for the same effects, using antioxidant activity as the dependent variable and analyzed by ANOVA with a 90% confidence level.

### 2.6. Anthocyanin Thermal Stability

The determination of the anthocyanin's thermal stability was performed for both best DES and methanolic extracts. The stability at 60 °C and 90 °C was performed as described by De Rosso and Mercadante [33], with modifications. After the extraction using the validated condition from the CCRD, the best DES and methanolic extracts were prepared and pH adjusted to 3.5 for all samples to keep anthocyanins stable (protonated). A food preservative (10 mg potassium sorbate) was added to both extracts to avoid spoilage. Then, each solution absorbance was adjusted to 0.8 through dilution, measured at the maximum absorption wavelength in the visible region for anthocyanins (520 nm) to standardize the assays, ensuring that Lambert–Beer law is considered and the citrate–phosphate buffer was utilized since small variations of pH affects anthocyanin concentration. The solutions were allowed to rest for 3 h to attain equilibrium among the different forms of anthocyanin. The solutions were then distributed in 10 mL centrifuge tubes with a total sample volume of 3 mL.

All tubes were placed in a water bath and heated at 60 °C or 90 °C, removing triplicates of each extract (DES or MeOH) at random times. The samples were immediately cooled in an ice bath for 10 min after removal. After this time, they were placed at room temperature for 50 min before reading the absorbance in a spectrophotometer (520 nm). When the absorbance pointed to half the original value (0.4), the assay was finished and results were plotted, and the emerging curve behavior was analyzed.

### 2.7. Anthocyanin Photostability

The anthocyanin's photostability was performed for both DES and methanolic extracts in the presence and absence of light. The extracts were prepared as described in Section 2.7.

The solutions were then distributed in 15 mL transparent tubes with a total sample volume of 10 mL.

The tubes were placed both in constant light and constant dark conditions, and samples of each triplicate of each extract (DES or MeOH) were removed at random times over the course of 25 days until absorbance reached half of its initial value (0.4). Samples had their absorbance were read in a spectrophotometer (520 nm) at these intervals. Results were plotted, and the emerging curve behavior was analyzed.

### 2.8. Disc-Diffusion Antimicrobial Sensitivity Test

The growth method was performed as follows: (1) three colonies, well isolated, of the same morphological type of *Salmonella enterica* subsp. (ATCC 13076) and *Staphylococcus aureus* (ATTCC 19095) were selected from the agar plate with Müeller–Hilton culture medium. Microorganisms were transferred to a tube containing 4–5 mL of culture medium. (2) The culture was incubated in broth at 35 °C until reaching or exceeding the standard McFarland 0.5 solution's turbidity measured in a spectrophotometer at 600 nm. (3) The turbidity of the growing culture was adjusted with a sterile saline solution in order to obtain optical turbidity comparable to that of the 0.5 McFarland standard solution. A sterile cotton swab was then dipped into the adjusted suspension up to 15 min after changing the inoculum suspension's turbidity. The dry surface of the Müeller–Hinton agar plate was inoculated by rubbing the swab across the agar's sterile surface ensuring uniform distribution of the inoculum. A set of antimicrobial discs was placed on the surface of a seeded agar plate, applying samples of the best DES extract, SF (saline), DES (deep eutectic solvent only), and positive control (vancomycin for *S. aureus* and meropenem for *S. enterica*), as well as samples of the optimized extract. The plates were inverted and placed in an oven at 35 °C for up to 15 min after applying the discs. After 16–18 h of incubation, each plate was examined. The diameters of the total inhibition halos were measured by the diameter (mm) of the disc.

## 3. Results and Discussion

### 3.1. Screening Process

A chromatographic profile of the methanolic jussara extract from DES screening analyzed the two major anthocyanins: C3G, with a retention time of 13.39 min, and C3R, with a retention time of 13.99 min (Figure S1 of Supporting Information). Quantification of DES and methanolic extract's peak areas to C3G and C3R, respectively, are shown in Figure 1. A confidence level of 95% was used for all analyses.

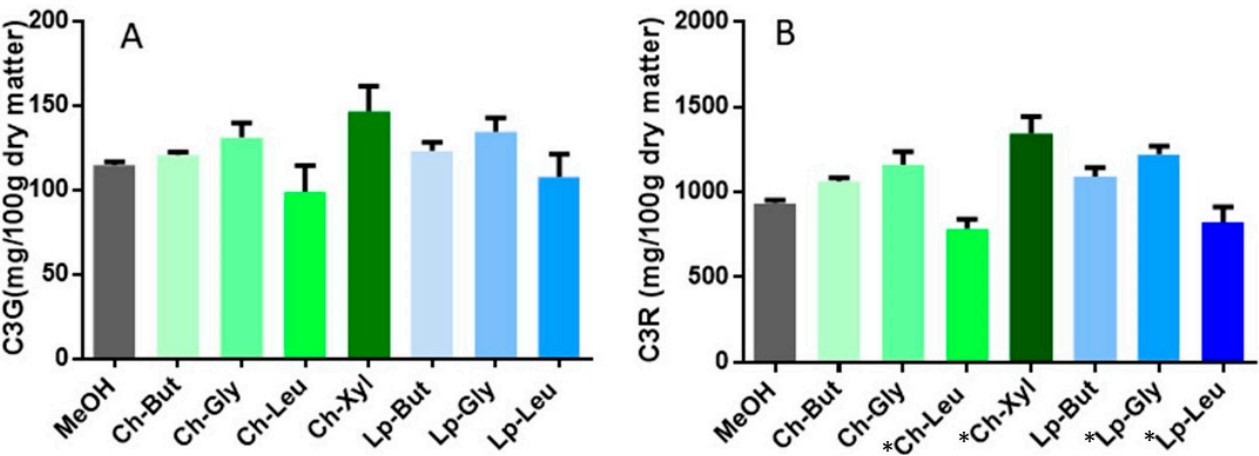

**Figure 1.** Cyanidin 3-glucoside (**A**) and Cyanidin 3-rutinoside (**B**) concentration (mg 100 g$^{-1}$ dry matter) in different extracts of eutectic solvents tested in lyophilized jussara samples. The * marks DES mixtures with results statistically different from the control (MeOH).

The results of C3G content in DES extracts were not significant (compared to MeOH; Mean: 114.79 ± 3.56 mg/100 g dry matter, ANOVA followed by Dunnett's test) for any of the extracts tested. The results of C3R in the extracts, on the other hand, were significant for Ch-Leu (785.16 ±99.2 mg/100 g dry matter, lower) and Lp-But (1092.41 ± 88.33 mg/100 g dry matter, lower), Ch-Xyl (1343.92 ±172.42 mg/100 g dry matter, higher), and Lp-Gly (1224.55 ± 80.20 mg/100 g dry matter, higher) in comparison to MeOH extract (935.48 ± 22.28 mg/100 g dry matter). C3R is the main anthocyanin of the pulp, and its yield significantly influenced the total anthocyanin level (TAL) of extracts when compared to methanolic extract.

After the sum of both anthocyanins, only the TAL of Ch-Xyl (1490.65 ± 198.27 mg/100 g dry matter) and Lp-Gly (1359.06 ± 92.92 mg/100 g dry matter) showed significant differences compared to the MeOH extract (1050.28 ± 29.32 mg/100 g dry matter). As our samples had 88.7% humidity, the TAL content in the methanolic extract can be expressed as 118.68 ± 3.3 mg/100 g fresh matter, being compatible with the MeOH extracts from the literature. One study [21] obtained 14.84 to 409.85 mg/100 g fresh matter with acidified methanolic extract, using samples gathered in different months. More recently [34], a change was observed in the total monomeric anthocyanins during *E. edulis* ripening. That study found that the anthocyanin content was very positively correlated with the ripening process, with a maximum of 634.26 mg TAL/100 g fresh matter. A progressive increase in the concentration of anthocyanins is believed to occur during ripening, and changes in the color of the fruits are easily observable [35]. Increased anthocyanin levels during ripening have also been previously noted for açaí (*Euterpe oleracea*) [35,36].

The medium absolute values of Ch-Xyl anthocyanin extracts were 42% higher than methanolic extract and Lp-Gly was 29% higher. These results clearly lead us to choose Ch-Xyl and Lp-Gly as the best green solvents for further experimental optimization of the extraction process (DES1 and DES2 of FFED, respectively).

### 3.2. Fractional Factorial Experimental Design (FFED)

After choosing the DES with a higher capacity to extract total anthocyanins from *E. edulis* lyophilized pulp, the optimization of different process conditions was conducted through a fractional factorial experimental design with three central points $2^{5-1}$, following the standardized effects described in the methodology section. The extraction process in different experimental conditions of the $2^{5-1}$ and the yields are shown in Table 4.

**Table 4.** Experimental values for C3G, C3R, and TAL obtained from fractional experimental design ($2^{5-1}$) with three central points for Ch-Xyl and Lp-Gly used as independent variables as well as $R_{(S/L)}$, extraction repetitions, time, and $R_{(CoSolvt\%)}$.

| Assay | Total C3G Content (mg/100g Dry) | Total C3R (mg/100 g Dry) | TAL (mg/100 g Dry) |
|---|---|---|---|
| | $Y_1 actual$ | $Y_2 actual$ | $Y_3 actual$ |
| 1 | 110.85 | 988.37 | 1099.21 |
| 2 | 81.82 | 747.47 | 829.29 |
| 3 | 101.52 | 933.65 | 1035.17 |
| 4 | 101.65 | 913.29 | 1014.93 |
| 5 | 133.67 | 1224.93 | 1358.60 |
| 6 | 116.76 | 1122.11 | 1238.87 |
| 7 | 118.17 | 1092.79 | 1210.96 |
| 8 | 110.04 | 997.73 | 1107.77 |
| 9 | 97.23 | 891.86 | 989.09 |
| 10 | 97.67 | 1059.43 | 1157.09 |
| 11 | 130.74 | 1264.24 | 1394.97 |
| 12 | 111.77 | 1063.85 | 1175.61 |
| 13 | 124.15 | 1174.53 | 1298.68 |
| 14 | 98.54 | 921.33 | 1019.87 |
| 15 | 100.84 | 961.75 | 1062.59 |
| 16 | 77.40 | 769.40 | 846.80 |
| 17 | 92.46 | 891.40 | 983.86 |

**Table 4.** *Cont.*

| Assay | Total C3G Content (mg/100g Dry) | Total C3R (mg/100 g Dry) | TAL (mg/100 g Dry) |
|---|---|---|---|
| **18** | 94.89 | 924.33 | 1019.22 |
| **19** | 95.94 | 908.96 | 1004.90 |

The experimental data shows that total anthocyanins extracted in all factorial design trials ranged from 829.29 to 1394.97 mg/100 g dry matter, indicating that the selected variables and levels could change the process efficiency up to 60%. As mentioned before, the proportion of DES is a qualitative variable and the levels studied were DES1 (−1) and DES2 (+1), and for the central point (0), we prepared a mixture of both DES. The negative effect of the proportion of DES, obtained from the FFED means that when we use DES2 instead of DES1, i.e., when we change the level from −1 to +1, the extraction of anthocyanins is less efficient. The fact that the DES2 variable has a negative effect implies that DES1 (Ch-Xyl) is more efficient in extracting the target bioactive compounds. Co-solvent usage had a positive influence, moving towards level +1 (more co-solvent in the total solvent used) and repetition (more repetitions). The solid/liquid ratio and time had no significant effects in this factional design. Considering these results, only two variables were carried out for the next step: co-solvent and repetitions.

Although the DES choice has had a significant effect, this variable could be fixed with Ch-Xyl for the CCRD ($2^2$) due to the negative mean effect of the Lp-Gly (DES2) solvent as discussed above. Chlorine chloride and xylitol were present in other tailor-made DES [37,38] for phenolic compounds and anthocyanins [39], where they are described as showing a high affinity for phenolic structure. Analysis obtained with the use of nuclear magnetic resonance spectroscopy by [40] shows that a Ch-Xyl–Water mixture shows great affinity for phenolic compounds compared to many other combinations of DES, including the Ch-Gly tested here. Additionally, Ch-Xyl and water exhibit a more cohesive structure when dissolving quercetin (a phenolic compound) than a citric acid/choline–chloride/water mixture [41]. However, in analyzing extraction of anthocyanins from mulberries, Ch-Gly gave a slightly better anthocyanin extraction than Ch-Xyl and a better extraction than mixtures of chloride–mannitol, chloride–fructose and choline chloride–glucose [39], indicating that food matrix and the type of anthocyanins contained can be important factors when tailoring DES for extraction of these compounds.

### 3.3. Central Composite Rotatable Design (CCRD)

As the literature indicates that using more than 50% of the water in the mixture can cause problems in DES stability [41,42], the co-solvent effect levels in CCRD were defined as 16% to 49.7%. For repetitions, previous experiments observed that beyond the fourth repetition, the solvent was almost colorless after extraction. A hypothesis was formulated then, assuming that after a hypothetical optimum of repetitions (four), more repetitions could generate TAL loss by excess manipulation. To verify this, a quick assay was done, with fixed variables (R(S/L) 1:15; Time: 2 min, co-solvent: 40%) with different repetitions (three, five, and seven times). Results are shown in Figure S2 of the Supporting Information, confirming a significant loss of efficiency in seven repetitions, compared to five repetitions (ANOVA).

After set repetition effect levels (2, 5 and 7 times), Table 5 shows the total yield obtained for the extraction process in different experimental conditions of the $2^2$ with central and axial points, for C3G, C3R, and TAL content from the *E. edulis* used as a response to the construction of the predictive model.

**Table 5.** Experimental values for C3G, C3R, and TAL extracted by CH-Xyl in the CCRD assays, and relative deviation (%) among experimental and predictive values of TAL.

| Assay | C3G (µg/mL) | C3R (µg/mL) | TAL (µg/mL) | Relative Deviation (%) |
|---|---|---|---|---|
| | $Y_1$ | $Y_2$ | $Y_3$ | $Y_3$ |
| 1 | 167.4752 | 1959.889 | 2127.364 | −3.38 |
| 2 | 142.7458 | 1804.298 | 1947.044 | −12.95 |
| 3 | 175.5200 | 2050.806 | 2226.326 | 20.31 |
| 4 | 182.2589 | 2016.332 | 2198.591 | 19.31 |
| 5 | 160.5043 | 1860.417 | 2020.922 | −6.15 |
| 6 | 165.3153 | 1826.223 | 1991.538 | −7.72 |
| 7 | 84.00578 | 1131.545 | 1215.551 | −27.34 |
| 8 | 156.6489 | 2012.959 | 2169.608 | 29.40 |
| 9 | 179.7469 | 2078.261 | 2258.008 | −0.83 |
| 10 | 186.2938 | 2071.040 | 2257.333 | −0.86 |
| 11 | 172.7000 | 2145.786 | 2318.486 | 1.80 |

The main effects and interactions were estimated for TAL are described in Equation (2).

$$\text{Total anthocyanin level } (\mu g/mL) = 2276.84 - 66.2\,(X_1)^2 - 212.6(X_2) - 224.0\,(X_2)^2 \quad (2)$$

where $X_1$ is co-solvent participation (%) and $X_2$ is the number of repetitions. ANOVA was used to evaluate the data's adequacy to verify the possibility of obtaining the model considering the different responses; additionally, the $R^2$ value provided a confidence measure of the model. In this study, an $R^2$ value of 0.6969 was obtained. The F value 94.9 (F value tabulated: 4.103) for TAL exceeded the 95% confidence level. The TAL response attended the criteria to generate the model and, thus, can be considered predictive; therefore, the response surface was plotted in Figure 2A.

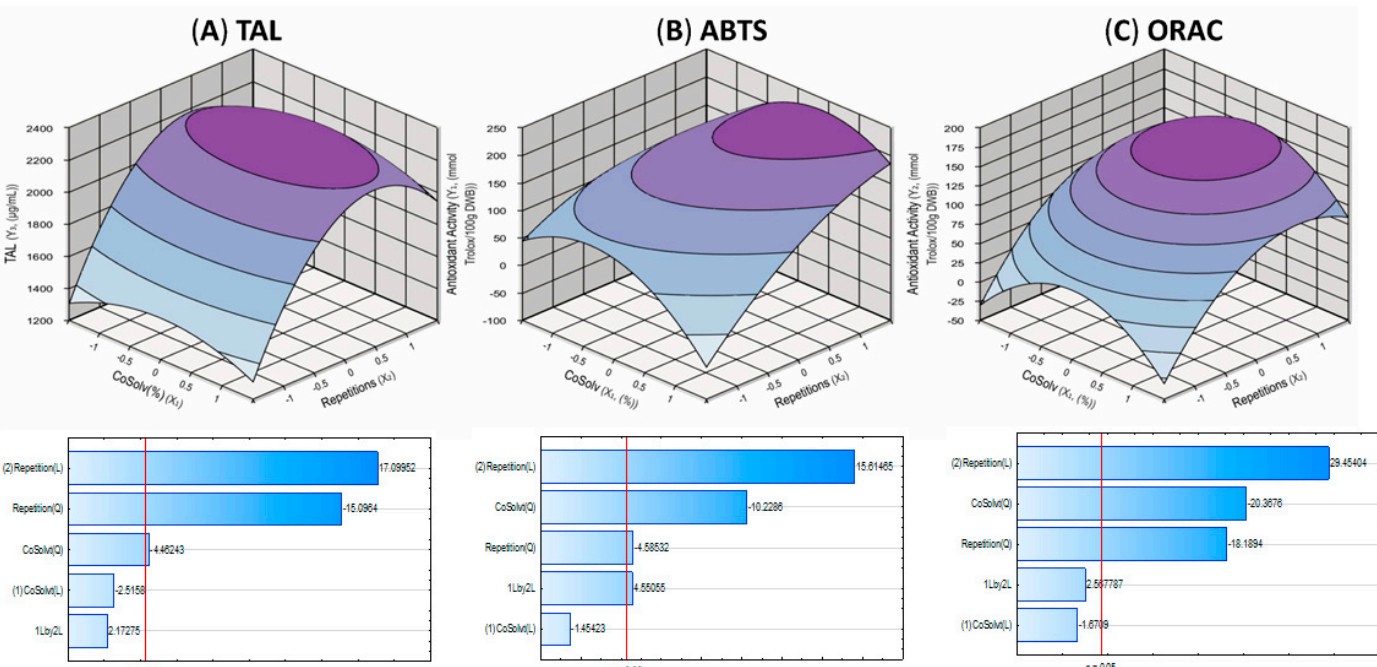

**Figure 2.** (**A**) Surface model for total anthocyanin level (TAL) (mg/100 g dry matter) from Ch-Xyl extract from *E. edulis* pulp for repetition and co-solvent as main effects and Pareto chart of the results. (**B**) Surface model for ABTS (mmol Trolox 100 g$^{-1}$ dry matter) and Pareto chart of the results. (**C**) Surface model for ORAC assay (mg/100 g dry matter) and Pareto chart of the results.

As observed in the model, repetitions largely influenced the TAL yield of extracts. On the other hand, next to maximum levels, the efficiency is lost; this is attributed to excess

manipulation of extracts. The co-solvent had a mild influence on the model, as observed in the Pareto chart of effects. The negative influence when the water participation (%) rises followed other studies, in which it is observed that it can disrupt the DES structure and thus reduce the DES effect in the extraction [25,43]. The best results were obtained around the central point level, with four repetitions and 30% co-solvent participation. The validation assays, performed in triplicate, were reproduced applying the best conditions. The model was effectively validated (mean of TAL was 2350.61 ± 53.05 μg/mL) with low relative deviation (3.10%), evidencing its high confidence in predicting data. These results for co-solvent participation are compatible with others in the literature for DES extraction for phenolic compounds that report best yields at 25–30% water usage [28,44]. Choline cations are highly capable of coordinating the surrounding environment: strong hydrogen bonding-mediated correlations between the hydroxyl group and water or chloride are believed to be formed. In addition, it is possible that the ammonium group drives the formation of a solvating environment, with water, chloride, and hydroxyl moieties approaching it, between the methyl groups [45]. The total anthocyanin content extracted by these assays varied from 1215.51 to 2318.48 μg/mL (assays 7 and 11, respectively), corresponding to 117.301 100 $g^{-1}$ fresh weight and 223.734 100 $g^{-1}$ fresh weight, respectively. Compared to the literature, a careful approach is necessary since the total anthocyanins contained in fruits can vary greatly due to edaphoclimatic variables and maturation stages [46]. The results obtained by [21] also using the experimental design ranged from 25.54 mg 100 $g^{-1}$ fresh weight (methanol) up to 418.52 mg 100 $g^{-1}$ fresh weight (Methanol/HCl 1.5 M). The study used an ultrasonic bath and the best condition with a 1:50 solvent ratio, which is considerably higher than that used in our study (1:15). In a study using the same solvent ratio [47] and 70% ethanolic solvent in ultrasonic bath, with 360 W potency, similar results were found: 190.92 ± 13.48 mg 100 $g^{-1}$ fresh weight to 282.64 ± 11.07 mg 100 $g^{-1}$ fresh weight. When authors tested a higher potency in the bath (900 W), a reduction in the anthocyanin content was observed.

### 3.4. Antioxidant Activity

After quantifying anthocyanin content in Ch-Xyl extracts for CCRD, the antioxidant activity of the samples was measured by ABTS and ORAC assays. Another CCRD assay was performed, using ABTS and ORAC assays as a response instead of TAL for the very same effects. Results are also shown in Figure 2B,C. Coded values can be found in Table S2A of Supporting Information.

In the ABTS assay, ANOVA resulted in an $R^2$ value of 0.88 and F value of 20.5, for a 95% confidence level; therefore, it is able to generate a model with a predictive equation (Equation (3)).

$$\textit{Antioxidant activity} \left(\text{mmol Trolox } 100/g\right) = 189.73 - 36.6 \left(X_1\right)^2 + 46.5(X_2) - 16.4 \left(X_2\right)^2 + 19.3 \left(X_1 \times X_2\right) \tag{3}$$

where $X_1$ is co-solvent participation (%), and $X_2$ is the number of repetitions. The graphical model and the Pareto chart of effects are shown in Figure 2B. For the ORAC assay (Table S2B of Supporting Information), ANOVA resulted in an $R^2$ value of 0.81 for a 95% confidence level and is therefore able to generate a model with a predictive Equation (4):

$$\textit{Antioxidant activity} \left(\text{mmol Trolox } 100/g\right) = 159.32 - 31.3 \left(X_1\right)^2 + 38.0(X_2) - 27.9 \left(X_2\right)^2 \tag{4}$$

where $X_1$ is co-solvent participation (%), and $X_2$ is the number of repetitions.

The precision and exactness of the model were validated by comparing the experimental results with the theoretical data predicted by Equations (2)–(4). As observed, response surfaces of the same extracts show slightly different performances for anthocyanin content and antioxidant activity. It is known that jussara pulp contains multiple hydrophilic compounds such as phenolic compounds [19,21,48,49], and even anthocyanin degradation products, which possess antioxidant properties [50]. Such compounds can explain the dissonant behavior and different maximum values of antioxidant activity when reacting

to ABTS and ORAC assays. The results had values between 69.92 and 198.93 for ABTS and between 63.09 and 196.436 mmol Trolox/100 g dwb (dry weight base) for ORAC. Values between 67.7 and 154.4 mmol Trolox/100 g dwb using TEAC (Trolox equivalent antioxidant capacity) and ORAC assays were found for jussara pulp [51]. While analyzing the ripening of the fruit, values between 108.8 (immature) and 207.1 (complete maturation) mmol Trolox/100 g dwb using the ORAC assay were found [52].Considering the evaluated dependent variables in CCRD (TAL and antioxidant activity by the methods ABTS and ORAC), the maximization of anthocyanin content and antioxidant activity are around the central point (applying around four repetitions and 30% co-solvent participation). This way, as presented for the TAL, the validation assays for antioxidant activity were also performed in triplicate for ABTS and ORAC. The result mean of the antioxidant activity was $198.82 \pm 10.97$ and $155.71 \pm 21.77$ (mmol Trolox $100\,g^{-1\,dwb)}$, representing a relative deviation from the predicted values of 5.06% and 10.51% for ABTS and ORAC, respectively. This confirms that the models were validated with low relative deviation, evidencing their statistical significance and predictiveness.

### 3.5. Determination of Anthocyanins Thermal Stability

The changes in anthocyanin concentration from jussara extracts caused by the temperature increase with the time was expected, following a biphasic behavior that was best fitted by a biexponential equation (Equation (5)), considering both evaluated temperatures (60 and 90 °C). Figure 3 shows the degradation kinetics of total anthocyanins extracted with MeOH and with Ch-Xyl, and the calculated kinetic parameters obtained by fitting the data from Equation (5) can be found in Table S3 of Supporting Information.

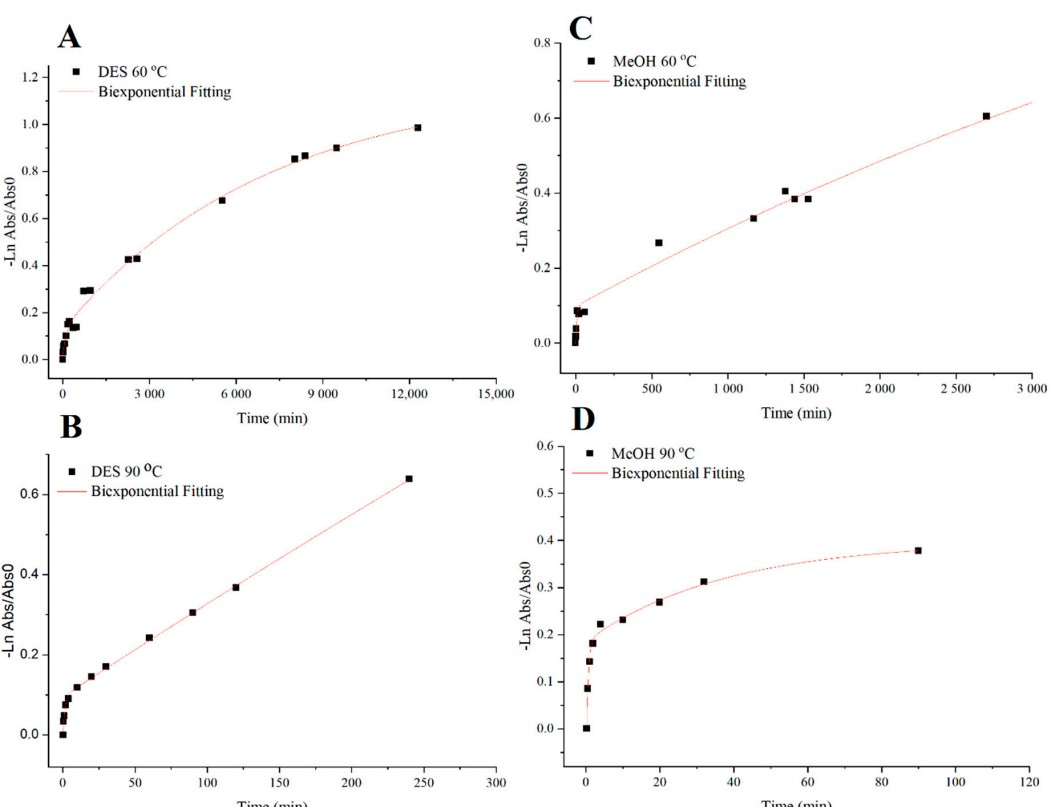

**Figure 3.** Degradation kinetics of anthocyanins extracted with Ch-Xyl (**A**,**B**) and MeOH (**C**,**D**) at 60 °C and 90 °C. The correlation coefficients of all curves are $R^2 \geq 0.99$.

The biexponential decay behavior, verified for the initial concentrations of anthocyanins, showed the same fast and slow lifetimes at each temperature, thus confirming the occurrence of a shared overall degradation mechanism.

$$y_t = A_1 exp(-\gamma_1 t) + A_2 exp(-\gamma_2 t) + y_\infty \tag{5}$$

$y_t$ and $y_\infty$ are anthocyanin absorbance values at real time and infinite time, respectively. $A_1$ and $A_2$ are the pre-exponential factors, whereas $\gamma_1$ and $\gamma_2$ are the observed rate constants for the fast and slow decays, respectively.

The degradation rate constants ($K_d$) at 60 °C and 90 °C from anthocyanins present in Ch-Xyl extract are shown in Table 6.

**Table 6.** Degradation constant ($K_d$) and half-life ($t_{1/2}$) values for anthocyanins extracted from jussara.

| Solvent | T (°C) | T (K) | $K_d$ (min$^{-1}$) | $t_{1/2}$ (min) | $E_a$ (kJ·mol$^{-1}$) |
|---------|--------|-------|-------------------|-----------------|----------------------|
| MeOH | 60 | 333.15 | 0.0002 | 3465.7 | 55.94 |
|  | 90 | 363.15 | 0.001 | 693.1 | |
| Ch-Xyl | 60 | 333.15 | 0.00008 | 8664.3 | 115.35 |
|  | 90 | 363.15 | 0.0025 | 277.3 | |

The $K_d$ values were estimated from the slope of the semi-natural logarithmic plot of residual activity vs. time, assuming classical Arrhenius linear behavior and using the equation between both temperatures. Similarly, the half-life ($t_{1/2}$) values were calculated using the degradation rate constants, also presented in Table 4. A comparison of the t$_{1/2}$ values of both extracts at different temperatures showed that the results of anthocyanins extracted with Ch-Xyl are more stable than those obtained with MeOH, with half-life 2.5 times greater, in both temperatures.

The activation energy ($E_a$) for the Ch-Xyl extract (115.35 kJ/mol) was higher compared to the MeOH extracts (53.94 kJ/mol), confirming that the Ch-Xyl utilization provides more thermally stable anthocyanins.

The literature shows that in several studies, the thermal degradation of bioactive compounds, such as enzymes and bioactive peptides, as well as other bioproducts, the kinetics followed a linear adjusted model. Meanwhile, the number of articles evaluating anthocyanins' stability using DES is limited, with different DES combinations and food matrices. Anthocyanins extracted from *Catharanthus roseus* with lactic acid-glucose (5:1) showed better stability in this DES than in acidified ethanol, about three times at 60 °C [9], although the curve showed a first-order model. Previous works from our research group showed that the thermal stability of carotenoids from tomatoes extracted using an ultrasound-assisted approach and 1-Butyl-3-methyl-imidazolium-chloride [BMIM][Cl] followed a first-order model [53]. However, they also showed the biexponential behavior in two studies regarding carotenoid extraction using distinct ionic liquids of orange peel [54] as well as from pupunha (*Bactris gasipaes*) fruit [55]. The higher stabilization ability of Ch-Xyl for anthocyanins may be correlated with the interactions between cyanidin and these molecules. The two components of Ch-Xyl may have intermolecular interactions, mainly hydrogen bonding with the carboxyl and hydroxyl groups of anthocyanins, as was observed for quercetin in Ch-Xyl [40]. This interaction is expected to decrease the movement of solute molecules, reducing its contact time with oxygen at the interface of anthocyanins and air, and consequently reducing oxidative degradation, which is the major degradation mechanism. In a study with anthocyanins extracted from black carrots using choline chloride/citric acid DES [56], the samples also exhibited improved stability.

### 3.6. Determination of Anthocyanins Photostability

Light is reported to have two different effects on anthocyanins: it accelerates the rate of the thermal reactions and the formation of new molecules, including chalcones [57]. Anthocyanins from jussara pulp changed with the time both in light and dark conditions

and followed a first order curve and were best fitted by a linear equation (Equation (6)), considering both situations. Figure 4 and Table 7 present the degradation kinetics of total anthocyanins extracted with MeOH and with Ch-Xyl.

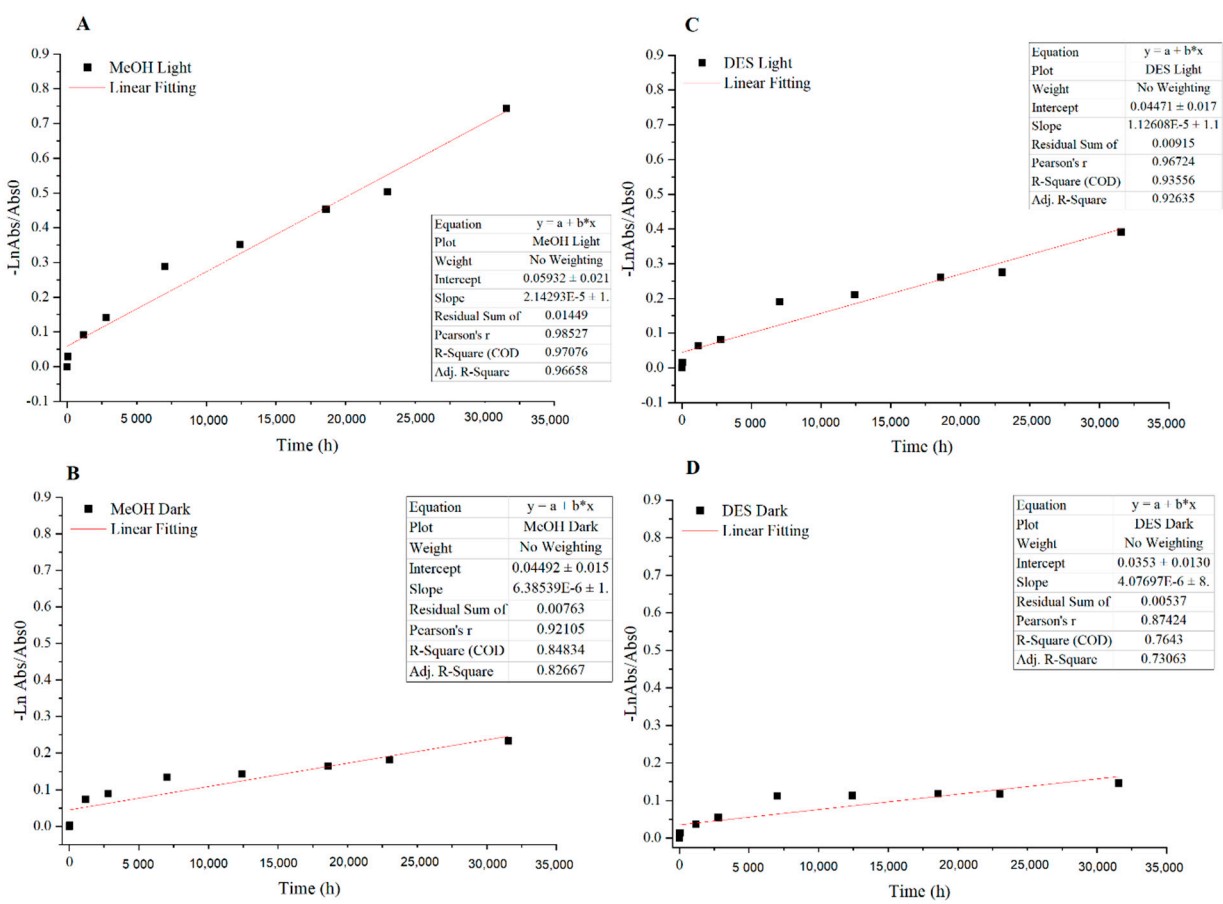

**Figure 4.** Degradation kinetics of anthocyanins extracted with MeOH (**A**) light and (**B**) Dark and Ch-Xyl (**C**) Light, (**D**) Dark in both conditions. The correlation coefficients of all curves are $R^2 \geq 0.99$.

**Table 7.** Degradation constant ($K_d$) and half-life ($t_{1/2}$) values for anthocyanins extracted from jussara.

| Solvent | Conditions | $K_d$ (h$^{-1}$) | $t_{1/2}$ (h) |
|---|---|---|---|
| MeOH | Light | 0.0015 | 462.1 |
| | Dark | 0.0005 | 1286.3 |
| Ch-Xyl | Light | 0.0008 | 866.4 |
| | Dark | 0.0003 | 2310.5 |

From a semi-natural logarithmic plot of the absorbance vs. time, the activation rate constants ($K_d$) were calculated, and the half-lives were estimated using Equation (6). The half-life ($t_{1/2}$) is defined as the time taken for the absorbance to reach 50% of the initial measurement.

$$t_{1/2} = \frac{-Ln0.5}{K_d} \qquad (6)$$

When comparing $t_{1/2}$ values of Ch-Xyl and MeOH, the latter perished significantly ($p < 0.05$, ANOVA) faster—about 1.9 times in dark and 1.5 in light—which indicates that Ch-Xyl extract could have a better storage time than MeOH extract, especially in dark conditions. Anthocyanins from blueberry extract exposed to natural and UV light had the same pattern of degradation after 25 days [58], while they retained about 60% of their initial

anthocyanin count in dark conditions and lower than 20% in light conditions. This study obtained 78% (MeOH) and 85% (Ch-Xyl) retention of initial absorbance after 25 days in dark and 43% (MeOH) and 63% (Ch-Xyl) retention of absorbance after 25 days, for jussara extracts exposed to light conditions. Degradation of anthocyanins of purple sweet potato analyzed for 15 days at 25 °C retained more than 80% anthocyanin content both under light and dark conditions [59]. The same was true for anthocyanins of *Hibiscus sabdariffa* for 10 days [60], and similar to results in this study, except for MeOH extract under light exposure, which retained only 63% of its initial absorbance after 15 days.

The strong hydrogen bond formed between Ch-Xyl and anthocyanins can once again explain the slower reaction rate of these extracts. This is pointed to as a possible strategy to deal with anthocyanins' relative instability compared to artificial colorants [39]. Evaluation of the thermal and light stability of anthocyanins from different sources is essential in order to determine their properties and hence apply those pigments, especially the most stable ones, for scaling-up processes and industrial applications. The bioaccessibility of anthocyanins during digestion, usually reported as very low [61], should be further evaluated when used in conjunction with Ch-Xyl solvents, evaluating their stabilization power in this environment to better understand the structure–stability relationships of the anthocyanin/extraction solvent.

### 3.7. Antimicrobial Activity

In order to evaluate the antimicrobial activity of extracts, a standard disk diffusion test for microbial sensitivity was conducted, and their inhibitory halo was measured (Figure 5 and Table S4 of Supporting Material).

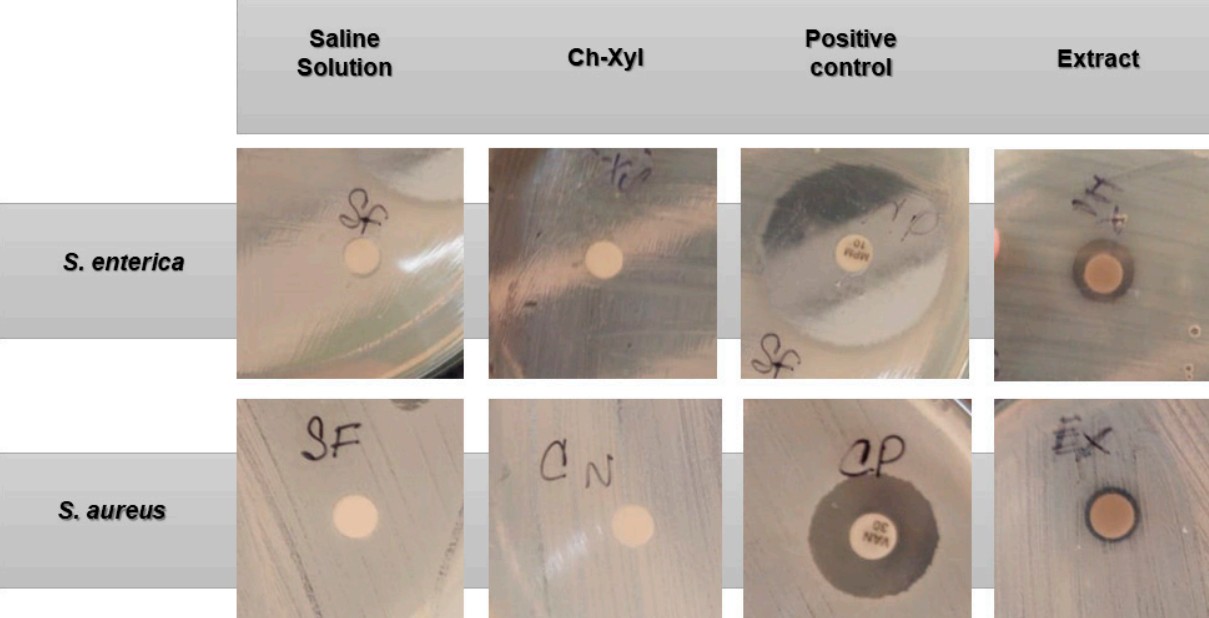

**Figure 5.** Disc-diffusion antimicrobial sensitivity. Representation of halos of inhibition for *S. enterica* and *S. aureus* in four treatments: saline solution, DES only (Ch-Xyl), positive control (meropen and vancomycin respectively), and optimized extract.

The bacterial pathogens, *Salmonella enterica* and *Staphylococcus aureus,* were tested, and the extract displayed moderate inhibition for both (8.6 mm and 7.6 mm halo respectively). *S. enterica* is one of the most frequent bacterial food-borne pathogens in humans. Salmonella infections range from gastrointestinal infections that are accompanied by inflammation of intestinal epithelia, diarrhea, and vomiting, to typhoid fever, a life-threatening systemic infection [62]. *S. aureus* is indeed found in the skin, hair, and nostrils of warm-blooded animals. This bacterium is a significant cause of nosocomial infections, as well

as community-acquired diseases. The extract resistance to this microorganism is a good indicator for stability and avoidance of microbial growth during storage. The results for *S. aureus* are according to [18], where *Euterpe edulis* crude extract displayed slightly positive inhibition against this microorganism. This can be attributed to the antimicrobial activity of anthocyanins and other polyphenols found in the extract, which is usually an antimicrobial agent in plants, and shows that DES extraction did not remove significant antimicrobial components from the crude extract [63]. Coupled with the relevant information of the stability of the extracts, further studies are necessary to determine the minimum inhibitory concentration (MIC) of the extract and to determine the ideal storage concentrations.

Together, these experiments build an additional step in the knowledge of how efficient and stable anthocyanin-rich extracts obtained with DES can be while keeping its antioxidant and antimicrobial properties. Further experiments should be conducted in order to investigate practical applications in a number of cosmetical and food products regarding the stability and interactions in these systems. Reducing the use of VOS is a worldwide effort, and general industry faces additional appeals for "green label" procedures by its consumers. Alternative green solvents fit this necessity well, and natural colorants obtained from jussara are a promising technology that may provide another sustainable economic option for the tropical forest remnants.

**Supplementary Materials:** The following supporting information can be downloaded at: https://www.mdpi.com/article/10.3390/pr10030615/s1, Table S1. (A) Real and coded values used in the fractional experimental design with three central composites (25-1). (B) Real and coded values used in the Central Composite Rotatable Design (CCRD) 22. Table S2. Real values and experimental values and relative deviation (%) among experimental and predictive values for (A) ABTS and (B) ORAC antioxidant activity for jussara pulp extracted by CH-Xyl in the CCRD assays. Table S3. Kinetic parameters obtained by fitting the experimental data for degradation of anthocyanins during thermal treatment of the extracts obtained with MeOH and DES. Table S4. Determination of halo diameter and susceptibility for extract for strains of *S. enterica* and *S. aureus*. S—sensible. Figure S1. Chromatogram obtained by HPLC-PDA-MS/MS of the jussara methanolic extract from jussara pulp. Cyanidin 3-glucoside, retention time: 13.396 min; cyanidin 3-rutinoside, retention time: 13.991. Figure S2. TAC of Ch-Xyl extracts by repetition. Purple dots represent individual experiments. Extracts with seven repetitions show significantly (ANOVA) less yield of anthocyanins than extracts with five repetitions. They are not statistically different of extracts with three repetitions.

**Author Contributions:** Conceptualization, V.V.D.R.; methodology, V.V.D.R. and A.R.C.B.; validation, V.V.D.R., A.R.C.B. and N.V.; formal analysis, N.V.; investigation, N.V., V.V.D.R. and A.R.C.B.; resources, V.V.D.R.; writing—original draft preparation, N.V., V.V.D.R. and A.R.C.B.; writing—review and editing, N.V., V.V.D.R. and A.R.C.B.; supervision, V.V.D.R.; project administration, V.V.D.R.; funding acquisition, V.V.D.R. All authors have read and agreed to the published version of the manuscript.

**Funding:** This work was supported by "Fundação de Amparo a Pesquisa do Estado de São Paulo-FAPESP" for fundings (2016/18910-1) and fellowship (2018/04265-2).

**Conflicts of Interest:** The authors declare no conflict of interest.

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
