# Peer review of "High-Performance Extraction Process of Anthocyanins from Jussara (Euterpe edulis) Using Deep Eutectic Solvents"

_processes, doi:10.3390/pr10030615_

Round 1

Reviewer 1 Report

 Dear Authors,

I consider the manuscript presented to me for review very good. The subject is interesting, the results obtained are valuable. The use of DES in the extraction of bioactive plants compounds  seems to be the optimal solution. In my opinion, the methodical approach is right, I appreciate the selection of DES components. In my opinion, it is important to evaluate the obtained extracts. I also appreciate the examination of the antibacterial properties of the obtained extracts. In my opinion, the research work was well and logically planned.The weaker side of this work is the discussion of the results. The authors could cite more literature research results. 

Author Response

I consider the manuscript presented to me for review very good. The subject is interesting, the results obtained are valuable. The use of DES in the extraction of bioactive plants compounds seems to be the optimal solution. In my opinion, the methodical approach is right, I appreciate the selection of DES components. In my opinion, it is important to evaluate the obtained extracts. I also appreciate the examination of the antibacterial properties of the obtained extracts. In my opinion, the research work was well and logically planned. The weaker side of this work is the discussion of the results. The authors could cite more literature research results.

R: Thank you for your review. The discussion part of the work was expanded in order to include more literature about the topic.

Reviewer 2 Report

The abstract and keywords should be reformulated according to the suggestions in the reviewed manuscript.

The paragraph 50-58 has no place here in the introduction. Maybe at conclusions. The introduction should be enriched with brief information on the jussara: systematic classification, biochemical composition, uses, ecological information. Other information on research to extract valuable compounds from this plant source and under what conditions, with what results. 

As you said in the abstract, the plant is an endangered species, found only in the Brazilian Atlantic Forest. How do you think it will be an adequate source of anthocyanins, when there are countless other more accessible plant sources? Argue the choice. Is there more research on the anthocyanin content in jussara? 

The entire chapter 2. Materials and methods must be corrected / completed / reformulated according to the instructions in the manuscript review.

Tables 2 and 3 should be divided so that the first columns, which are useful for understanding the experimental design, should be found in that subchapter (2.4.), and only the columns with results should be kept in their current location. 

Figures 3 and 4 must be completed with A and B according to each explanation.

Subchapter 3.8. needs to be reformulated. In our opinion, all extracts should be evaluated comparatively, against several microorganisms, in order to draw a conclusion regarding the antimicrobial activity. 

Author Response

Reviewer 2

The abstract and keywords should be reformulated according to the suggestions in the reviewed manuscript.

R: thank you for your suggestion, the abstract was reformulated.

The paragraph 50-58 has no place here in the introduction. Maybe at conclusions. The introduction should be enriched with brief information on the jussara: systematic classification, biochemical composition, uses, ecological information. Other information on research to extract valuable compounds from this plant source and under what conditions, with what results.

R: Thank you for your suggestion. The paragraph (50-58 lines) was removed. Brief information about the relevant topics were added in the introduction.

As you said in the abstract, the plant is an endangered species, found only in the Brazilian Atlantic Forest. How do you think it will be an adequate source of anthocyanins, when there are countless other more accessible plant sources? Argue the choice. Is there more research on the anthocyanin content in jussara? 

R: Thank you for highlight this point. The text was modified to clarify the range of occurrence (from north east parts of Brazil coast down to eastern Paraguay and northern Argentina). Despite the habitat fragmentation and over exploration of the heart of the palm, important populations of the palm still occur, where the local communities make use of the fruit and the occasional (illegal due conservation status) heart of the palm extraction. Namely Vale do Itajaí, Vale do Ribeira and Ubatuba. Most of the effort done by environmentalists and ecologists that study the plant argue that improving the use and economic potential of the fruit is the best alternative to discourage heart of the palm extraction. In this sense, many most studies in last decade aim to evaluate biological compounds (Jamar et al., 2020; Rocha et al., 2018; A. Santamarina et al., 2018; A. B. Santamarina et al., 2018; Schulz et al., 2015, 2017; Silva et al., 2014) of the fruit and methods of extraction (Garcia-Mendoza et al., 2017; Lima et al., 2012; Passos et al., 2015) also extensively revised by our group (Vannuchi et al., 2021). This study develops a green extraction method for the anthocyanins of the fruit using deep eutectic solvents for the first time, with potential use as colorant and food additive, that once employed can increase the fruit pulp demand, and therefore encourage its sustainable exploration.

The entire chapter 2. Materials and methods must be corrected / completed / reformulated according to the instructions in the manuscript review.

R: Thank you for your suggestions. Entire chapter 2 was reformulated to better comprehension. All other correction were done. Discussing particular points:

Line 171 – Anthocyanin content can by influenced by pH: At pH 1, the cation flavylium is predominant, while at pH values between 2 and 4, the quinoidal species are more common. At pH values between 5 and 6, a carbinol pseudobase and chalcones rise in concentration (Castañeda-Ovando et al., 2009). Subtle changes in pH could otherwise lead to inconsistencies between replicas and therefore pH 3.5 was fixed with buffer. Additionally, potassium sorbate 10mg was added to avoid spoilage that could eventually occur (more likely in photostability assays at room temperature but it was added in all stability assays to avoid biases.

Line 179 – 10 days was not previously chosen. Samples were removed until absorbance reached half initial absorbance. This was achieved after 10 days for samples at 60º C and 6h for 90º C samples. Text was modified to be clearer at this point.

Line 180 and 190 – Times chosen to evaluate the assays were randomly determined as need. Since At 90º C the decay is relatively fast, 15s, 30s, 45s and 60s were needed to obtain the curve slope. For 60º C, 2 min, 4min, 10min were enough. For photostability assays even longer (after 1 or 2 days). For example, MeOH extracts at 60º C: Almost no change in absorbance was noted between 20min and 60min, so another one was taken only after 550 min as shown in the table below. Random times continued to be chosen until absorbance of 0.4 (or lower) was obtained in order to better visualize the the half-life time.

Time (min)

ABS

0

0,8462

2

0,831616667

4

0,81468

10

0,7765

20

0,782733333

60

0,7793

550

0,648133333

1170

0,6074

1380

0,564866667

1440

0,576466667

1530

0,576766667

2700

0,4624

2850

0,458043333

2910

0,447633333

2970

0,4599

5760

0,31722

Line 451 – Ch-Xyl (not ChCl only) was the best solvent mixture for anthocyanin extraction (evaluated after CCRD) and therefore all stability, antioxidant and microbiological tests were conducted only with the extract of this solvent.

Tables 2 and 3 should be divided so that the first columns, which are useful for understanding the experimental design, should be found in that subchapter (2.4.), and only the columns with results should be kept in their current location. 

R: Thank you for your suggestion. This modification was done.

Figures 3 and 4 must be completed with A and B according to each explanation.

R: Thank you for your suggestion. This modification was done.

Subchapter 3.8. needs to be reformulated. In our opinion, all extracts should be evaluated comparatively, against several microorganisms, in order to draw a conclusion regarding the antimicrobial activity. 

R: Thank you for your suggestion. With the changes in the material and methods section we believe that is clear now that only one extract (Ch-Xyl) was chosen for further experiments after CCRD, because it wielded more anthocyanins than other extracts.

Reviewer 3 Report

The manuscript entitle “High-performance extraction process of anthocyanins from Jussara” includes the study and comparison of different DES and using water as co-solvent in order to extract the maximum quantity of anthocyanins from the pulp of Euterpe edulis Mart. To accomplish this, the authors carry out an experimental design, with different extraction conditions to optimize the maximum extraction of the aforementioned compounds. Furthermore, the results were compare with methanolic extracts.

The study is interesting, however, at some points it is somewhat confusing and lacks information. Below I write a series of questions/suggestions to try to improve this.

Page 2. Section Fruit gathering. The authors do not specify the season and the year of harvest of the fruit,  this could be interesting to later compare the results with publications of this fruit in different phenological stages.

Page 2. “… the fruit was depulped, macerated and inmediately frozen at -40ºC, lyophilized at 48 h, and then stored at -40ºC to preserve the anthocyanin content.” Why do the authors say that the fruit was macerated and in what solvent?

The macerated fruit was frozen at -40ºC, lyophilized at 48 h? explain this part better because it is not understood, it was frozen and then lyophilized for 48 hours?

Why here has the fruit been frozen at -40ºC and then the DES at -80ºC?

Later the authors talk about the moisture content, how has it been determined?

Page 2. Section 2.2. The authors write that "the DES was mixed and subsequently frozen at -80ºC and then lyophilized to remove water". Although the DES were synthesized by the heating method, authors do not explain how the mixture was carried out, e.g. temperatures taken for each mixture, photos of the mixtures would also be appreciated. Authors also say that the water is removed from the mixture. You do this in order to know exactly the amount of co-solvent added? the authors speak of 7 mixtures but then I count 8. In addition, the authors say that these mixtures already exist in the literature, could they then explain the novelty in this part of the work? only its application for the extraction of anthocyanins in this particular fruit?

Page 3. Screening of seven eutectic solvents on the anthocyanin extaction. Why have the authors chosen those extraction conditions (400 W) using ultrasound-assisted extraction? Is there optimization or can you cite some previous work?

In this part, the explanation of the choice of the optimum DES is confusing. Authors do this part to choose the non-quantitative variable, that is, the choice of the best DES? if so, specify it.

Authors literally say "After compared to methanolic extraction..." Compare what? by the previous sentence, it could be understood that the comparison is with the content in anthocyanins, but it is better to specify it again.

Page 3, section 2.4. Experimental design to….”(i) eutectic solvent composition. What does this mean? It is also confusing that here the authors put both the variables that are going to be used in the design and the fixed variables. Authors can separate both variables and mark only the variables of the design in sections. The design part for me is quite confusing because the authors speak of fixed variables such as number of repetitions (3 times), time of extraction (5 min), when in the table there are several values both for the repetition (1, 3 and 5) and for the extraction time (2, 4 and 6). By the way, with repetitions the authors mean extraction repetition? In addition, the table of supplementary table 1S.A does not indicate the 19 trials that are indicated in Table 2...

Authors write that the cosolvent was fixed at 30% when in the table I see different values.

After performing the extraction, the extracts were filtered. Could you indicate the filters used?

Page 4, 2.5. Identification and quantification of anthocyanins. Could you describe the linear gradient used in another, clearer way?

Page 4, 2,6. Antioxidant activity. Reduce the theoretical part of ORAC and leave it similar to ABTS.

Page 5. 2.7. Anthocyanin thermal stability. 10 mm potassium sorbate. Are the units correct?

Page 6. “ in figure 1.a. to C3G and figure 1.b.to C3R”.

Better rewrite the sentences "Results of C3G content in DES extracts..." and “Being C3R….of extracts compared to methanolic extract”.

Figure 1. Indicate the statistics.

Paragraph after figure 1. What conclusions do we draw from the results, and from what is observed in the literature? What state of maturation did the fruits of this study have to compare results with literature?

Rewrite the sentences “A progressive  increase…., and a phase of decreasing concentration”, and “The medium absolute values of Ch-Xyl….”.

Page 7. Explain better what is DES1 and DES2.

Page 7. Why DES1 is more efficient extracting the target compounds? Discuss this results with literature.

Page 8. “… attributed to the DES but to its individual compounds and water.”Explain this better.

Page of table 4, number 12? what does [BMIM][Cl] mean? in table 4 which solvent is DES?

Figure 5. Improve photo resolution.

Conclusions section. Last paragraph of page 5 and first of 6 put it as a separate section of conclusions.

Author Response

Reviewer 3

The manuscript entitle “High-performance extraction process of anthocyanins from Jussara” includes the study and comparison of different DES and using water as co-solvent in order to extract the maximum quantity of anthocyanins from the pulp of Euterpe edulis Mart. To accomplish this, the authors carry out an experimental design, with different extraction conditions to optimize the maximum extraction of the aforementioned compounds. Furthermore, the results were compare with methanolic extracts.

The study is interesting, however, at some points it is somewhat confusing and lacks information. Below I write a series of questions/suggestions to try to improve this.

Page 2. Section Fruit gathering. The authors do not specify the season and the year of harvest of the fruit, this could be interesting to later compare the results with publications of this fruit in different phenological stages.

R: Thank you for your suggestion, it was added.

Page 2. “… the fruit was depulped, macerated and immediately frozen at -40ºC, lyophilized at 48 h, and then stored at -40ºC to preserve the anthocyanin content.” Why do the authors say that the fruit was macerated and in what solvent.

R: Thank you for your note. This section was changed, the fruit was macerated in the depulper at same time the peel and seed is removed, as a traditional process for production of jussara and açaí (the same). No solvent was used.

The macerated fruit was frozen at -40ºC, lyophilized at 48 h? explain this part better because it is not understood, it was frozen and then lyophilized for 48 hours?

R: Thank you for your note. Yes, they were frozen then lyophilized during 2 days in the freeze dryer

Why here has the fruit been frozen at -40ºC and then the DES at -80ºC?

R: Thank you for your question. They were both frozen at -40º C. This was changed in the text.

Later the authors talk about the moisture content, how has it been determined?

R: Thank you for your question. The moisture content was performed according to the methods of AOAC, added in the text.

Page 2. Section 2.2. The authors write that "the DES was mixed and subsequently frozen at -80ºC and then lyophilized to remove water". Although the DES were synthesized by the heating method, authors do not explain how the mixture was carried out, e.g. temperatures taken for each mixture, photos of the mixtures would also be appreciated. Authors also say that the water is removed from the mixture. You do this in order to know exactly the amount of co-solvent added? the authors speak of 7 mixtures but then I count 8. In addition, the authors say that these mixtures already exist in the literature, could they then explain the novelty in this part of the work? only its application for the extraction of anthocyanins in this particular fruit?

R: Thank you for your questions. Only 7: [Ch-Leu 1:2], [Ch-But 1:2], [Ch-Gly 1:2], [Ch-Xyl 5:2], [Lp-Leu 1;2], [Lp-But 2:5] and [Lp-Gly 2:5]. Yes, during the heating, water is added to the mixture, so the lyophilization is carried in order to remove it more effectively (to be sure of the 30% co-solvent content of water added later). The solvents exist in the literature but some of them not used for anthocyanins or this fruit specifically, though all are used to phenolic compounds. We don’t have photos of all the DES mixtures since they were used more than 2 years ago, in the beginning of the process. Nevertheless, we can manage to re-acquire them and synthetize to take photos if needed.

Page 3. Screening of seven eutectic solvents on the anthocyanin extraction. Why have the authors chosen those extraction conditions (400 W) using ultrasound-assisted extraction? Is there optimization or can you cite some previous work?

R: Thank you for your question. The potency of use of the probe was tested in previous works of the group and with 400W the best results were observed. Therefore, this was not a variable of this work.

In this part, the explanation of the choice of the optimum DES is confusing. Authors do this part to choose the non-quantitative variable, that is, the choice of the best DES? if so, specify it.

R: Thank you for your question. The DES choice was made considering total anthocyanin content of the samples. A initial screening of the 7 DES was made with fixed variables to comparison. The 2 best DES were chosen for the fractional factorial experimental design (FFED) (LP-Gly, Ch-Xyl). They were both then used for a second experimental design, central composite rotatable (CCRD). One of the variables was the solvent: Level +1 Ch-Xyl, Level -1 LP-Gly, and Level 0 a mix of both of them, 50/50. The text was modified to made it clearer.

Authors literally say "After compared to methanolic extraction..." Compare what? by the previous sentence, it could be understood that the comparison is with the content in anthocyanins, but it is better to specify it again.

R: Thank you for your suggestion. Text was modified.

Page 3, section 2.4. Experimental design to….”(i) eutectic solvent composition. What does this mean? It is also confusing that here the authors put both the variables that are going to be used in the design and the fixed variables. Authors can separate both variables and mark only the variables of the design in sections. The design part for me is quite confusing because the authors speak of fixed variables such as number of repetitions (3 times), time of extraction (5 min), when in the table there are several values both for the repetition (1, 3 and 5) and for the extraction time (2, 4 and 6). By the way, with repetitions the authors mean extraction repetition? In addition, the table of supplementary table 1S.A does not indicate the 19 trials that are indicated in Table 2...

R: Thank you for your question. Text of methodological part was modified to be clearer. The first experimental design (FFED) goal was to remove non-significant variables. The CCRD was done with number of repetitions (v) and co-solvent content (%) were chosen as significant variables, the rest was fixed (as they were non-significant for FFED).

Authors write that the cosolvent was fixed at 30% when in the table I see different values.

R: Thank you for your question. Co-solvent was fixed for the initial screening of the 7 DES. For FFED and CCRD it was a variable. Text was modified to be clearer.

After performing the extraction, the extracts were filtered. Could you indicate the filters used?

 R: Thank you for your question. It was used filter paper under vacuum, this information was added to the text.

Page 4, 2.5. Identification and quantification of anthocyanins. Could you describe the linear gradient used in another, clearer way?

R: Than you for your suggestion. Text was modified to be clearer.

Page 4, 2,6. Antioxidant activity. Reduce the theoretical part of ORAC and leave it similar to ABTS.

R: Thank you for your suggestion, the description was reduced and reference added.

Page 5. 2.7. Anthocyanin thermal stability. 10 mm potassium sorbate. Are the units correct?

 R: Thank you for your question. No, it was 10 mg.

Page 6. “ in figure 1.a. to C3G and figure 1.b.to C3R”.

Better rewrite the sentences "Results of C3G content in DES extracts..." and “Being C3R….of extracts compared to methanolic extract”.

R: Thank you for your suggestion it was modified.

Figure 1. Indicate the statistics.

R: Thank you for your suggestion. A * was added to show significant variables.

Paragraph after figure 1. What conclusions do we draw from the results, and from what is observed in the literature? What state of maturation did the fruits of this study have to compare results with literature?

 R: Thank you for your question. The obtained the fruits from IPEMA institute, that handle sustainable areas for E. edulis reforestation. They use fruits in the later stages of maturation, where they show more anthocyanin content.

Rewrite the sentences “A progressive  increase…., and a phase of decreasing concentration”, and “The medium absolute values of Ch-Xyl….”.

 R: Thank you for your suggestion, it was modified.

Page 7. Explain better what is DES1 and DES2.

R: Thank you for your suggestion, it was modified in the methods section and the result.

Page 7. Why DES1 is more efficient extracting the target compounds? Discuss these results

with literature.

 R: Thank you for your suggestion. More literature comparison was added to the text.

Page 8. “… attributed to the DES but to its individual compounds and water.”Explain this better.

R: Thank you for your question. DES solvents in a solution with too much water can be considered dissolved in water, leading to disappearance of all the signals of the hydroxyl groups in NMR, what is believed happen due to the complete rupture of the hydrogen bonds, as observed to 1,2-propanediol and choline chloride with more than 50% water (Dai et al., 2015). Also,  as observed in this article (López et al., 2020), NMR results choline-chloride/xylitol/water mixture form a supramolecular structure between xylitol, choline chloride and water molecules, that disappears when high water content is added.

Page of table 4, number 12? what does [BMIM][Cl] mean? in table 4 which solvent is DES?

 R: Thank you for your question. In table 4 only Ch-Xyl solvent was used. Methanol was used only in the screening phase as control. We believe that changes in the methodology section made this understanding clearer. 1-Butyl-3-methyl-imidazolium-chloride name was added for [BMIM][Cl].

Figure 5. Improve photo resolution.

 R: Thank you for your suggestion we applied another format.

Conclusions section. Last paragraph of page 5 and first of 6 put it as a separate section of conclusions.

R: Thank you for your suggestion, the text was modified to better understanding.

Reviewer 4 Report

The manuscript describes the extraction process of anthocyanins from Jus- 2
sara. Here are a few comments that can improve the manuscript. 

  1. Abstract needs to be improved.
  2. The application of RSM in determining optimized extraction conditions should be explained in abstract.
  3. The authors need to show that the species of the plant has been identified by the expert.
  4. In 2.4 (methodology), is it possible to evaporate DES using rotary evaporator?
  5. Figure 1, mark (asterisk ofr small letter) should be put on the graph bar to show statistically significant differences between variables.

I believe the manuscript is suitable for publication after minor modification. 

Author Response

Reviewer 4

The manuscript describes the extraction process of anthocyanins from Jus- 2
sara. Here are a few comments that can improve the manuscript. 

  1. Abstract needs to be improved.

R: Thank you for your suggestion, abstract was changed.

  1. The application of RSM in determining optimized extraction conditions should be explained in abstract.

R: Thank you for your suggestion, abstract was changed.

  1. The authors need to show that the species of the plant has been identified by the expert.

R: Thank you for your suggestion. We obtained the pulp from Instituto de Permacultura (IPEMA) that description of fruit gathering was obtained from them. This information was added.

  1. In 2.4 (methodology), is it possible to evaporate DES using rotary evaporator?

R: Thank you for your observation. No, it’s not possible, rotary evaporator is used to remove water added as co-solvent to concentrate the extract.

  1. Figure 1, mark (asterisk ofr small letter) should be put on the graph bar to show statistically significant differences between variables.

R: Thank you for your suggestion. The modification was done.

Round 2

Reviewer 2 Report

I believe that the authors have made the necessary improvements to the manuscript as much as possible and agree with the publication of the article.

Author Response

Reviewer 2

I believe that the authors have made the necessary improvements to the manuscript as much as possible and agree with the publication of the article.

R: Thank you for evaluating our work.

Thank you for evaluating our work.

Reviewer 3 Report

The authors have made a great effort to considerably improve the text according to the indications raised in the first revision, however I would like to indicate some points that have not been modified and that in my opinion continue to be confusing, and therefore, I believe that their modification would help further improve understanding of the research.

Page 5, line 120. the authors have included the fusion points in the table, but making the mixtures do not specify the working temperatures, can you specify it?

Page 5, lines 123-124. The authors wrote 8 DES in the text, including  [Lp-Xyl 2:3]. So, include 8 DES in text or delete this DES if it was not included in the investigation.

Page 6, lines 143, 148 and tables 1 and 2. I reiterate what do the authors mean by "repetitions"? extraction repetitions? if so, include it in the text and tables.

Page 8, line 151. With filter paper authors mean using whatman quantitative paper filter? specify the characteristics and commercial brand of the paper filters.

Table 4, 5 and corresponding text. I reiterate in eliminating DES and putting directly the solvent Ch-Xyl. It is confusing to use such a generic term when a specific one is already being used .

Author Response

Reviewer 3

The authors have made a great effort to considerably improve the text according to the indications raised in the first revision, however I would like to indicate some points that have not been modified and that in my opinion continue to be confusing, and therefore, I believe that their modification would help further improve understanding of the research.

Page 5, line 120. the authors have included the fusion points in the table, but making the mixtures do not specify the working temperatures, can you specify it?

R: Thank you for your question. All mixtures were done by heating at 80ºC. The fusion points were showed to illustrate that they are usually higher (like chorine chloride 247ºC), but the deep eutectic capacity of the mixtures have much lower fusion points. As the fusion points of the individual components were not needed to be understanding this paper and it is possible to generate this confusion, so we remove them after your observation.

Page 5, lines 123-124. The authors wrote 8 DES in the text, including [Lp-Xyl 2:3]. So, include 8 DES in text or delete this DES if it was not included in the investigation.

R: Thank you for your note. It was removed.

Page 6, lines 143, 148 and tables 1 and 2. I reiterate what do the authors mean by "repetitions"? extraction repetitions? if so, include it in the text and tables.

R: Thank you for you note, it was modified to extraction repetitions.

Page 8, line 151. With filter paper authors mean using whatman quantitative paper filter? specify the characteristics and commercial brand of the paper filters.

R: Thank you for your note. It was Unifil qualitative paper 80g/m2.

Table 4, 5 and corresponding text. I reiterate in eliminating DES and putting directly the solvent Ch-Xyl. It is confusing to use such a generic term when a specific one is already being used.

R: Thank you for your suggestion. Several instances of DES 1 were excluded in favor to Ch-Xyl.